# Tiny Green Army: Fighting Malaria with Plants and Nanotechnology

**DOI:** 10.3390/pharmaceutics16060699

**Published:** 2024-05-23

**Authors:** Isabelle Moraes-de-Souza, Bianca P. T. de Moraes, Adriana R. Silva, Stela R. Ferrarini, Cassiano F. Gonçalves-de-Albuquerque

**Affiliations:** 1Immunopharmacology Laboratory, Department of Physiological Sciences, Federal University of the State of Rio de Janeiro—UNIRIO, Rio de Janeiro 20211-010, Brazil; isabelle.moraes2@gmail.com (I.M.-d.-S.); biancapt@gmail.com (B.P.T.d.M.); 2Immunopharmacology Laboratory, Oswaldo Cruz Foundation, FIOCRUZ, Rio de Janeiro 21040-361, Brazil; arsilva71@gmail.com; 3Pharmaceutical Nanotechnology Laboratory, Federal University of Mato Grosso of Sinop Campus—UFMT, Cuiabá 78550-728, Brazil; stela.ferrarini@ufmt.br

**Keywords:** green nanotechnology, plant extracts, *Anopheles*, *Plasmodium*, ACT

## Abstract

Malaria poses a global threat to human health, with millions of cases and thousands of deaths each year, mainly affecting developing countries in tropical and subtropical regions. Malaria’s causative agent is *Plasmodium* species, generally transmitted in the hematophagous act of female *Anopheles* sp. mosquitoes. The main approaches to fighting malaria are eliminating the parasite through drug treatments and preventing transmission with vector control. However, vector and parasite resistance to current strategies set a challenge. In response to the loss of drug efficacy and the environmental impact of pesticides, the focus shifted to the search for biocompatible products that could be antimalarial. Plant derivatives have a millennial application in traditional medicine, including the treatment of malaria, and show toxic effects towards the parasite and the mosquito, aside from being accessible and affordable. Its disadvantage lies in the type of administration because green chemical compounds rapidly degrade. The nanoformulation of these compounds can improve bioavailability, solubility, and efficacy. Thus, the nanotechnology-based development of plant products represents a relevant tool in the fight against malaria. We aim to review the effects of nanoparticles synthesized with plant extracts on *Anopheles* and *Plasmodium* while outlining the nanotechnology green synthesis and current malaria prevention strategies.

## 1. Introduction

In 2021, there were a total of 247 million cases of malaria, causing 619 thousand deaths, with a mortality rate of 14.8 per 100,000 population at risk. Compared to the previous year, there was an increase of 2 million malaria cases globally [1]. Changes in malaria epidemiology have been attributed to increased risk of transmission, parasite genetic variability, growing drug resistance, and the COVID-19 pandemic [2]. In the first year of the pandemic, malaria combat was compromised because of the reduced distribution of Insecticide-Treated Nets (ITNs) and the disruption of treatment and diagnosis of malaria cases, especially in the World Health Organization (WHO) African Region [1]. Due to the pandemic, the lockdown limited the action of malaria prevention programs, causing a reduction in the distribution of ITNs. Also, health systems were overwhelmed with a shortage of health workers due to a lack of protection equipment, the closure of clinics due to quarantines, and all efforts being directed to control the pandemic, directly impacting the management and prevention of new malaria cases [3]. The WHO reported that in 2021, there was an increase in malaria cases, mainly in the African and Asian continents, where 268 million people required assistance due to humanitarian crises, such as floods, hunger, and political conflicts [1].

Malaria is an infectious disease caused by the *Plasmodium* sp. parasite and transmitted by female *Anopheles* sp. mosquitoes during blood feeding. Around forty *Anopheles* species are known to be human malaria vectors, e.g., *A. stephensi*, *A. gambiae*, *A. coluzzii*, *A. funestus* [1,4,5]. Anophelinae is from predominantly tropical locations because warmer temperatures are optimal for its survival and development [6]. Gravid females lay eggs in still bodies of water, which hatch into larvae. Larvae turn to pupae after four stages (instars) of growth. Pupae is the resting stage where feeding stops, and the adult mosquito emerges above water after a few days. In the adult stage, female mosquitoes feed from blood to develop eggs [7] and can acquire the malaria parasite if it feeds from an infected host.

*Plasmodium* has an intricate life cycle, needing two different hosts to complete its development. The parasite enters the bloodstream at human infection and reaches the liver, where it matures. Then, the parasite is released back into the blood, invading erythrocytes and initiating sexual reproduction [8], generating gametes that can be transmitted to mosquitoes.

According to the WHO guidelines for malaria [9], case management involves early diagnosis and prompt treatment, assuring correct dosing, rational medicine use, and drug choice. The WHO established artemisinin-based combination therapy (ACT) as the primary treatment for most malaria cases. The treatment uses artemisinin and its derivatives, such as artemether, dihydroartemisinin, and artesunate, in combination with other drugs [9]. The ACT therapeutic scheme is necessary due to the emergence of artemisinin-resistant parasites [10,11]. Genomic studies enabled the identification and tracking of genetic mutations in the parasite (reference Chaves et al., 2022, for a better review on this topic) [12]. The proposed mechanisms of *Plasmodium* resistance are the effluxion or blocked entry of the drug through mutations in membrane transporters and the enhancement of metabolic processes to protect from drug-mediated protein damage [13,14]. Resistance to first-line and easily accessible antimalarial drugs, such as chloroquine, quinine, and mefloquine, directly impacts the increase in disease cases in some tropical countries, threatening the local health systems.

The immunity that patients develop after contact with the parasite is not permanent. So, individuals become susceptible again over time. The vast majority of antimalarials used in the treatment of malaria present toxicity, with commonly reported gastrointestinal adverse effects, like nausea and vomiting. Which makes it difficult for the patient to adhere to the treatment, compromising its efficacy [15]. Antimalarial treatment aims to eliminate all forms of plasmodia that produce clinical signs. Different drugs are used for this, depending on the parasite species, the patient’s age, and the disease’s severity.

While vector control strategies focus on using insecticides, resistance to these chemicals has also been observed, posing another threat to malaria transmission control. As an adaptation to ITNs, mosquitoes can change their feeding behavior to ensure survival [16]. Also, the frequent use of insecticides causes genetic modifications that improve the mosquito detoxification system, which allows for lower susceptibility to insecticides-based interventions [17,18,19].

Hence, despite being treatable and preventable, malaria management is complex because of the parasite’s resistance to current drug treatments, mosquitoes’ resistance to insecticides, and the environmental impacts of the latter. In light of these difficulties, there is a relevant need to improve research on novel treatments, drug targets, vector control, and the consequent development of better technology to deliver these strategies.

## 2. Green-Nanotechnology

Nanotechnology involves manipulating matter in atomic or molecular scales to generate new products and materials, with sizes ranging from 1 to 100 nanometers [20]. Nano systems can be classified as organic, produced from lipids or polymers widely used in the pharmaceutical industry, or inorganic, which are metal-based. Nano systems differ in terms of composition, the method of preparation, and structural organization [21]. The most common methods for synthesizing nanoparticles (NPs) utilize expensive and toxic substances as reducing agents, such as sodium borohydride, hydrazine, formaldehyde, and hydrogen [22]. So, to overcome toxicity, several nanocarriers are being evaluated for developing these particles, such as plant extracts, for a biogenic synthesis approach. Nanotechnology has numerous biomedical applications in wound healing [23], cancer treatment [24,25,26], dentistry [27], and the cosmetic industry [28]. Also, a clinical trial evaluated the effects of thyme and carvacrol NPs on *Aspergillus fumigatus* isolated from patients in intensive care (found in clicaltrial.gov with the ID NCT04431804).

Because of economic benefits, easy accessibility, and variety, plant extracts are a relevant alternative [29]. NPs synthesized with plant extracts have appropriate size and shape, are consistent with average NPs, and are efficient because of high yield in less synthesis time [30]. Also, using plant extracts requires the bottom-up method of NPs synthesis, characterized by the self-assembly of atoms and molecules from chemical interactions, which is simpler and more efficient [20]. A very commonly used metal in the synthesis of NPs, silver (Ag), has antimicrobial and anti-inflammatory effects [29]. Thus, using plant extracts to synthesize Ag NPs can enhance these activities because polyphenols also show beneficial properties [31]. Moreover, Ag NPs are harmful to aquatic organisms and are considered a pollutant. However, the green synthesis of these NPs can reduce Ag toxicity by increasing molecule stability and reducing the agglomeration of particles [30]. Although these advantages are significant, some drawbacks must be considered, such as possible toxicity, the lack of biocompatibility of the materials used, and the high cost of obtaining nano systems compared to conventional pharmaceutical formulations to treat a neglected disease [12].

The green synthesis of NPs refers to using biological materials as the capping agents to improve the form, functionality, and stability of NPs [32]. The process of green-synthesizing NPs is illustrated in Figure 1.

Capping agents refer to molecules with an amphiphilic character, featuring a polar head and a non-polar tail region. The polar head group forms coordination bonds with the metal atoms within nanocrystals formed during agglomeration, while the tail region interacts with the surrounding medium. The right reducing and capping agents used during the synthesis of NPs are necessary to ensure optimal activity. The bioactive compounds of plant-based products are currently underestimated due to degradation and volatilization in field conditions. One option to avoid these drawbacks is to formulate bioactive plant products using polymers, plasticizers, stabilizers, and biodegradable antioxidants [33]. A plant extract is composed of a variety of metabolites and reductive biomolecules that play a crucial role in reducing metal ions. These components include terpenoids, flavones, ketones, aldehydes, amides, carboxylic acids, carbohydrates, proteins, and vitamins [34]. During the process of green synthesis, bio-compounds form a coating on the surface of NPs [32]. This supplementary layer consequently amplifies the biological attributes of green NPs in contrast to those generated through other chemical reduction methods. The conventional green synthesis reaction involves a straightforward combination of a natural extract and a metal salt solution, serving as the precursor for NPs [35]. This strategy has garnered attention over the past decade, especially for Ag and Au NPs, which are considered safer than other types of metallic NPs [36]. Other metals, like nickel (Ni), manganese (Mn), Titanium (Ti), titanium trichloride (TiO_2_), palladium (Pd), cerium (Ce), platinum (Pt), and ZnO, have been utilized in the creation of plant-based NPs and green NPs synthesis [35,36].

Biomolecules exhibit the capacity to transfer electrons to metallic ions, functioning as potent reducing agents that initiate the ions’ reduction. Consequently, the reduced ions develop structured patterns that resemble a crystalline formation called a nucleus. The nucleation process begins with the monomer aggregation in supersaturated systems. The formation of an atomic nuclei is followed by the expansion of these nuclei into metal nanoparticles [37]. These nuclei offer a foundation for the reduced ions, facilitating their gradual deposition onto the surface, thereby contributing to the enlargement of the particles. The expansion process is halted by biomolecules that can also function as capping agents, binding to the particle surface and effectively stabilizing their size [37,38].

Besides metallic NPs, the discussion extends to nano emulsions and nanogels. Nanogels are hydrogel particles composed of crosslinked polymers that can swell in an aqueous environment in nanoscale size. Notably, nanogels can accommodate a substantial drug load and achieve efficient encapsulation. Also, they excel at encapsulating larger quantities of substances and sustaining the release of entrapped molecules [39]. On the other hand, nano emulsions are nanostructures formed by the dispersion of two immiscible liquids, such as oil in water or water in oil, that are stabilized by a surfactant. It has a variety of dosage forms, including liquids, creams, sprays, gels, aerosols, and foams. The pesticide industry uses it as an aqueous base for organic deliverables [40]. The characteristics of NPs directly influence their interaction with the environment. Hence, they should be thoroughly investigated to prevent any adverse effects. The small size of NPs and the negative charge on their surface were associated with the potential of Ag NPs to cause hemolysis [41].

The application of nanotechnology for both vector control and patient therapy represents the most promising solution to this challenge. The main objective of malaria treatment is to provide the ideal drug concentration for the intracellular parasite [42]. Nanotechnology protects drugs from chemical degradation, increases the drug’s circulation time in the blood, improves cellular interaction with the parasite, and promotes the maintenance of plasma levels at a constant concentration. Also, it increases therapeutic efficacy, the progressive and controlled release of the drug, decreases toxicity by reducing plasma peaks at maximum concentration, and adds to the possibility of directly affecting specific targets [43]. Nanoparticles can be functionalized with ligands that specifically bind to receptors on the surfaces of malaria parasites. This targeted drug delivery ensures that anti-malarial drugs are delivered directly to the parasites, increasing treatment efficacy while minimizing side effects on healthy cells. Also, nanoparticles can serve as carriers for vaccine antigens, enhancing their stability and immunogenicity [44].

## 3. Plants and Nano Formulations

In their natural habitat, plants already show defense mechanisms against plagues, predators, and other competing species [33,45]. Thus, to use those against a naturally occurring mosquito is to take full advantage of the biological power displayed by these plants in favor of human health whilst causing minimal environmental impact.

Especially in the malaria scenario, plants that grow in tropical and subtropical regions can facilitate green insecticide production because of easy access, rare toxicity against humans, and low-dose effectiveness [46]. For example, extracts of various seaweed species, especially *Caulerpa acemose*, are toxic to mosquito larvae, including *A. stephensi*, at the half-lethal concentration (LC50) of 0.06 µg/mL [47]. *Zingiber cernuum* essential oil is an effective larvicide and oviposition deterrent against malaria vectors *A. stephensi* and *A. subpictus* and does not affect non-target mosquito predators [48]. *Heracleum sprengelianum* essential oil is also effective against *A. subpictus* larvae with LC50 33.4 µg/mL [49].

Notably, diverse plant species are commonly used in traditional medicine practices to treat malaria and alleviate Campo’s symptoms [50] because these plants show antiparasitic properties, as seen in [51,52,53,54,55]. Historically, plants have originated the active substances used in malaria drug treatments, such as quinolines derived from *Cinchona* trees, which originate chloroquine [56]. Also, the current drug of choice to treat malaria, artemisinin, is derived from the plant *Artemisia annua* L. Its discovery can be traced back to the sixties when it was used as an herbal medicine in China [57].

The challenge in using crude plant extracts is the degradation and evaporation of the chemical compounds upon field conditions [33]. In addition, most isolated plant-derived substances have a short half-life [58], emphasizing the need to develop a better form of administration. Phytochemical nanoformulation (i.e., non-nutrients derived from plants) can improve bioavailability, water solubility, and protection from enzymatic degradation [59]. Botanical-based NPs can be an eco-friendly alternative to current chemical pesticides, which are highly toxic to the environment [60].

Apart from plants, many other biological sources have been used to synthesize NPs. Nano-pesticide synthesized using *Metarhizium robertsii* showed larvicidal activity against *A. stephensi* fourth instar larvae in low doses and was not toxic to non-target organisms, such as earthworms and crustacea [61]. Sardine-fish-scale Ag NPs showed potent larvicidal, pupicidal, and ovicidal activity against *A. stephensi*. Interestingly, the study showed increased predatory activity against larvae from the mosquitofish *Gambusia affinis* [62]. Ag NPs synthesized by *Bacillus marisflavi* showed great ovicidal, larvicidal, and pupicidal activity against *A. stephensi*, the mortality rates of eggs and pupae being the highest compared to other mosquito species [63]. Chitosan Ag NPs showed toxicity against *A. stephensi* larvae (fourth instar LC50 5.51 ppm) and pupae (LC50 6.54 ppm) [64]. Mycosynthesized Ag NPs using *Cochliobolus lunatus* were toxic to second–fourth instar larvae of *A. stephensi* at concentrations of 5 and 10 ppm while presenting no danger to non-target *Poecilia reticulata* [65]. Earthworms’ coelomic fluids have been extensively used due to their therapeutic properties, such as cytotoxic, proteolytic, and antibacterial. Researchers used the *Eudrilus eugeniae* earthworms as reducing and stabilizing agents in synthesizing Ag NPs tested against *A. stephensi*. *E. eugeniae* Ag NPs were effective against larvae and pupae, and the predation efficiency of the mosquitofish *G. affinis* was boosted from 68.50% to 89.25% in second instar larvae [66]. *G. affinis* and *P. reticulata*, among other species, are larvivorous fish commonly used in the biological control of *Anopheles* mosquito by malaria prevention programs [67] and demonstrated to be highly effective in a malaria-free country [68].

Moreover, nanoformulated natural products have proven effective against other disease-carrying mosquitoes, such as *Aedes* sp. [69,70,71,72,73,74,75,76,77], and *Culex* sp. [78,79,80]. Also, it shows antimicrobial [64,81,82,83], and acaricidal [84] effects.

Additionally, nanotechnology can be applied to other strategies to fight malaria, such as better drug delivery in malaria cases, as reviewed by our group [12], and the nanoformulation of repellents to control disease transmission [85].

## 4. Malaria Prevention Strategies

Whereas targeting the mosquito vector, the main combat strategies are Indoor Residual Spraying (IRS) and ITNs [9]. In sub-Saharan Africa, ITNs are the primary prevention strategy, and even though there has been an increase in ownership and use of ITNs over the years, population-level access has decreased since 2017. In addition, the levels of people protected by IRS in endemic countries have decreased, reaching 2.4% in 2021 compared to 5.5% in 2010 [1]. Although IRS and ITNs have effectively reduced the malaria burden, these strategies have faced challenges because of household accessibility [86] and vector resistance to insecticides [87]. Pyrethroid-only long-lasting insecticidal nets (LLINs) are the current main WHO recommendation, but mosquito resistance to pyrethroids has compromised its efficacy. Pyrethroids are the synthetic version of pyrethrins derived from the flowers of the Pyrethrum genus, which have insecticidal effects. The development of pyrethroids was incited due to the molecular instability of pyrethrins [88]. Pyrethroids increase the opening of insects’ sodium channels, disrupting nervous impulse propagation and the overactivation of nerve cells [89], causing mosquito paralysis, the same effect observed in nanoformulated plant extracts and further discussed in this review. Other synthetic insecticides are proposed with conditional recommendation because cost-effectiveness is still uncertain and lacks evidence [9]. Pesticides are known to be hazardous to the environment [90]. They are lethal to aquatic organisms [91,92] and potentially affect human development due to prenatal exposure [93] and hormone signaling dysfunction [94].

Currently, the WHO does not recommend insecticide space-spraying due to a lack of evidence about its impact on malaria and the short life of the used chemicals [9]. This review summarizes the effects of numerous eco-friendly alternatives to synthetic pesticides, emphasizing nanotechnology’s use to improve efficacy (Table 1). *Senna occidentalis* and *Ocimum basilicum* coils showed significant toxicity against *A. stephensi*, causing mortality rates of 38% and 52%, respectively, similar to the pyrethrin-based control coil, which caused a 42% mortality rate [95]. *Pteridium aquilinum* Ag NPs coils synthesized with different plant parts caused mortality rates comparable to pyrethrin-based coils [96]. Also, in both studies, the plant-containing smoke could reduce parasite transmission because it caused a reduction in the percentage of fed mosquitoes. Coils from *Ulva lactuca* were effective against *A. stephensi*, causing a mortality rate of 66%, higher than that of the pyrethrin-based coil, which was 41% [97].

In addition, the green nanotechnology strategy can promote a biological form of vector control because most plant extracts can increase the predatory activity of non-target organisms against the early developmental stages of the mosquito. For instance, chitosan-synthesized Ag NPs increased the predation efficacy of *Danio rerio* against larvae of *A. stephensi* at a concentration of 1 ppm [64]. *Citrus limon* gold–palladium (Au-Pd) NPs had no negative impact on the predatory habit of flying insects, which increased over time [98]. Likewise, *Lagenaria siceraria* zinc oxide (ZnO) NPs increased *P. reticulata* predation efficacy from 45.8% in the aqueous extract to 61.13% [99]. Also, *Cymbopogon citratus* Au NPs boosted *Mesocyclops aspericornis* predation against *A. stephensi* from 26.8 to 45.6% [100]. The predation efficacy improvement could be due to the decreased mobility of larvae, which are typically fast-paced, after exposure to the green-synthesized NPs [98]. Additionally, Ag NPs synthesized with *Ichnocarpus frutescensi* [101], *Rubus ellipticus* [102], *Naregamia alata* [103], and *Hugonia mystax* [104] had a negligible toxic effect on non-target aquatic organisms, such as *G. affinis*, *Diplonychus indicus*, and *Anisops bouvieri*. *Solanum xanthocarpum*-synthesized Ag NPs [105] and *Momordica charantia* leaf extract TiO_2_ NPs [106] had no toxicity against *P. reticulata* at the same concentrations lethal to mosquitos. It is crucial to outline the choice of material used to synthesize NPs because CdS showed toxicity against non-target aquatic organisms [107]. These are predators of mosquito breeding sites, hence the importance of developing insecticides harmful only to the intended species to maintain the ecologic niche balance. *Malva sylvestris* Ag NPs were not toxic to *D. indicus* and *G. affinis*, as the longevity and swimming habits did not change after exposure to NPs, and the suitability index showed that treatment with NPs was more toxic to the mosquito larvae [108].

The studies above follow the recently defined WHO preferred product characteristics (PPCs) for vector-control products. PPCs were determined to incentivize the search and development of new strategies to control *Anopheles* mosquitoes while addressing multiple parameters, including indoor and outdoor transmission control, complementarity to existing methods, and safety for humans and the environment [87]. Green-synthesized nanotechnology fulfills most of the demanded PCCs, thereby being a suitable candidate to fight malaria.
pharmaceutics-16-00699-t001_Table 1Table 1Characteristics of NPs synthesized with plant extracts that have beneficial use and its effects on malaria vector and parasite. The nanoparticle sizes are according to transmission electron microscopy (TEM) or scanning electron microscopy (SEM) analysis.Plant ExtractPlant UseType of NanotechnologySize of NanoparticlesEffectsReferences*Alchornea cordifolia* leafInfections, inflammation, rheumatism, analgesia, and arthritisAg NPs5–25 nm(TEM)Larvicidal, impact on larvae behavior and morphology, antiplasmodial and hemolytic[109]*Aristolochia indica* leafPost–partum infections, snakebites, fever, malaria, skin diseases, helminths, edema, intestinal disorders, and antibacterialAg NPs30–55 nm(SEM)Larvicidal and pupicidal in laboratory and field, predation booster[110]*Azadirachta indica* leaf and barkAntimalarial, intestinal disorders, and diabetesAg NPs4–28 nm(TEM)Antiplasmodial and hemolytic[41]*Cedrus deodara* oilInsecticidalPectin nanocapsules40–80 nm (TEM)Larvicidal and impact on larvae behavior and morphology[111]*Cocos nucifera* mesocarpFoodAg NPsmean 23 nm (TEM)Larvicidal [112]*Couroupita guianensis* flowerAntibiotic, antifungal, antidepressant, antiseptic, analgesia, stomach ache, and skin diseasesAu NPs29.2–43.8 nm(TEM)Larvicidal and pupicidal in laboratory and field, adulticidal, predation booster, and antiplasmodial[113]*Codium tomentosum*AntioxidantAg NPs20–40 nm(SEM)Larvicidal, pupicidal, antiplasmodial, antioxidant, antibacterial, and predation booster[114]*Eclipta prostrata* leafLipidemia, atherosclerosis, hepatic diseases, and snake-venom poisoningPd NPs18–64 nm(TEM)Antiplasmodial and cytotoxic[84]*Mimusops elengi* leafCardiotonic, stomach ache, anti-helminthic, dysentery, antimicrobial, anti-ulcer, antianxiety, anti-oxidant, antihyperglycemic, anti-hyperlipidemic, anti-inflammatory, and feverAg NPs25–40 nm (TEM)Larvicidal, pupicidal, adulticidal, and predation booster in laboratory and field[115]*Arachis hypogaea* peelCattle foodAg NPs20–50 nm (TEM)Larvicidal and impact on larvae morphology[116]*Vitex negundo* leafBactericidal, diabetes, inflammation, and asthmaZnO NPs28–42 nm(TEM)Larvicidal, pupicidal, antioxidant, cytotoxic, and photocatalytic[117]*Pteridium aquilinum* leafAnalgesia and foodAg NPs35–65 nm(SEM)Larvicidal and pupicidal in laboratory and field, adulticidal, ovicidal, antiplasmodial, and repellent smoke[96]*Ulva lactuca*Antioxidant, antibacterial, and antiviralAg NPs20–35 nm(SEM)Larvicidal, pupicidal, antiplasmodial, and repellent smoke[97]*Citrus limon* leafNatural pesticide, insect repellent, and antimicrobialAu-Pd NPs1.5–18.5 nm(TEM)Larvicidal and predation booster[98]*Lagenaria siceraria* peelAnti-swelling, diuretic, antibacterial, and cytotoxicZnO NPs-Larvicidal, impact on larvae behavior and morphology, antiplasmodial, predation booster, and cytotoxic[99]*Cymbopogon citratus* leaf-Au NPs20–50 nm(TEM)Larvicidal, pupicidal, and predation booster[100]*Ichnocarpus frutescens* leafAnti-diabetes, anti-tumor, anti-inflammatory, and analgesiaAg NPs5–47 nm(TEM)Larvicidal and biocompatible with non-target organisms[101]*Rubus ellipticus* leafDiabetes, diarrhea, gastralgia, wound healing, dysentery, antifertility, antimicrobial, analgesia, and epilepsyAg NPs1–25 nm(TEM)Larvicidal, ovicidal, and adulticidal, oviposition deterrent, and biocompatible with non-target organisms[102]*Naregamia alata* leafWound healing, ulcers, halitosis, cough, asthma, bronchitis, splenomegaly, scabies, pruritus, dysentery, dyspepsia, catarrh, anemia, and malarial feversAg NPs0–5.5 nm(TEM)Larvicidal, ovicidal, adulticidal, and biocompatible with non-target organisms[103]*Hugonia mystax* leafAnthelmintic and rheumatismAg NPs40–90 nm(SEM)Larvicidal and biocompatible with non-target organisms[104]*Solanum xanthocarpum* leafAnti-cancer, antioxidant, anti-HIV, antibacterial, and insecticidalAg NPs10–20 nm(TEM)Larvicidal and biocompatible with non-target organisms[105]*Eucalyptus globulus* oilNatural pesticideNanoemulsion22–40 nm(TEM)Larvicidal in laboratory and semi-field[118]*Mangifera indica* leafAntioxidant and antibacterialTiO_2_ NPs30 nm(TEM)Larvicidal and acaricidal[84]*Barleria cristata* leafAntioxidant, cytotoxic, and antimicrobialAg NPs38–41 nm(TEM)Larvicidal, biocompatible with non-target organisms[119]*Malva sylvestris* leafAntioxidant, anti–inflammatory, and antimicrobialAg NPs18–25 nm(TEM)Larvicidal and biocompatible with non-target organisms[108]*Ammania baccifera* aerialAnalgesia, antifertility, hypothermic, hypertensive, antiurolithiasis, antisteroidogenic,antimicrobial, antiurolithic, anti–inflammatory, and antioxidantAg NPs10–30 nm(TEM)Larvicidal[120]*Artemisia nilagirica* leafAntimicrobial, anthelmintic, antiseptic, and larvicidalAg NPs<30 nm(SEM)Larvicidal and pupicidal[121]*Valoniopsis pachynema*-CdS NPs<100 nm(SEM)Larvicidal, pupicidal, antiplasmodial, and toxic to non–target organisms[107]*Vitex negundo* leafAntimicrobial, anti–inflammatory, diabetes, asthma, cytotoxic, and larvicidalZnO NPs28.48–42.14 nm(TEM)Larvicidal and antioxidant[117]*Annona squamosa* leafPesticidal, cytotoxic, and antioxidantAg NPs200–500 nm(SEM)Larvicidal, pupicidal, ovicidal, and adulticidal[122]*Momordica charantia* leafAntidiabetic, antiviral, antitumor, antileukemic, antibacterial, anthelmintic, antimutagenic, antimycobacterial, antioxidant, antiulcer, anti-inflammatory, hypocholesterolemic, hypotriglyceridemic, hypotensive, immunostimulant, and insecticidalZnO NPs21.32 nm(SEM)Larvicidal, acaricidal, and pediculicidal[123]

TiO_2_ NPs70 nm(TEM)Larvicidal, pupicidal, antiplasmodial, and biocompatible with non–target organisms[106]*Euphorbia hirta* leafNatural insecticideAg NPs30–60 nm(SEM)Larvicidal and pupicidal[124]*Nerium oleander* leafAnticancer, antimicrobial, anxiolytic, and antipsychoticAg NPs20–35 nm(SEM)Larvicidal and pupicidal[125]*Heliotropium indicum* leafFever, throat infection, ulcer, gonorrhea, localizedinflammation, rheumatism, ring worm, wounds, aphrodisiac, astringent, and expectorantAg NPs18–45 nm(TEM)Adulticidal[126]*Zornia diphylla* leafDysentery, venereal diseases, and sleep inductionAg NPs30–60 nm(SEM)Larvicidal and biocompatible with non-target organisms[127]*Mussaenda glabra* leaf-Ag NPs15–25 nm(TEM)Larvicidal and biocompatible with non-target organisms[128]*Anisomeles indica* leafInflammatory skin diseases, liver protection, intestinal infections,abdominal pain and immune system deficiencies, expectorant, diaphoretic, rheumatism, and psoriasisAg NPs18–35 nm(TEM)Larvicidal[129]*Holostemma adakodien* leaf-Ag NPs-Larvicidal, antibacterial, and biocompatible with non-target organisms[81]*Quisqualis indica* leafAntimicrobial and anticoccidialAg NPs<30 nm(SEM)Larvicidal and biocompatible with non-target organisms[130]*Nicandra physalode* leafAntidiuretic,mydriasis, analgesia, antibacterial, anti-inflammatory, and insecticidalAg NPs5–35 nm(SEM)Larvicidal and biocompatible with non-target organisms[131]*Gmelina asiatica* leafHepatic diseasesAg NPs20–64 nm(TEM)Larvicidal[132]


## 5. Plant Extracts and Green NPs Action Mechanisms

The NPs’ action mechanisms are still unclear, and the current knowledge is limited to mosquitoes’ morphological, molecular, biochemical, and physiological levels [133]. Table 1 summarizes the effects of plant extracts in these numerous levels with the nanoparticle’s corresponding type and size. The high mortality rates exerted by NPs on mosquito larvae may be explained by the small size of the particles, which enables the passage through the respiratory tubes and/or insect cuticle and into cells where they interfere with molting and other physiological processes [66,122]. After penetrating the exoskeleton, the Ag NPs exert their effects in the intracellular space, interacting directly with proteins or DNA and causing genotoxic effects. The depletion of antioxidants induces oxidative stress, the oxidative dissolution of Ag NPs, and mitochondria perturbation [134]. It also can induce apoptosis through the activation of procaspase-3, the downregulation of the pro-survival protein Bcl-2, the enhanced expression of pro-apoptotic gene products, and the release of cytochrome c into the cytosol [135,136].

The toxicity of NPs is intricately linked to their propensity to accumulate within various organs. Upon entering the bloodstream, NPs exhibit the capacity to disperse throughout the body, accumulating within organs such as the liver, spleen, lungs, and kidneys. This distribution of NPs within the body is primarily influenced by their surface area-to-size ratio, a factor that affects their tendency to accumulate in distinct tissues and organs [137]. The pathways through which NPs exert toxicity within the organs go through several critical factors, including producing reactive oxygen species (ROS), DNA damage, the alteration of protein structures and functions, and compromising membrane integrity. Notably, NP characteristics that seem to enhance these mechanisms involve their expansive surface areas, which promote molecular interactions within the specific target sites [138]. Toxicity is not limited to the target species but also to the environment, especially in the management of the vector in-field conditions. NPs released in water affect aquatic organisms, consequently resulting in reports of blood acidosis, causing circulatory collapse, bioaccumulation, hepatotoxic effects, oxidative stress, and embryonic development. However, such effects were observed at elevated concentrations [29], in opposition to the low concentrations used in nanoparticles.

The neurotoxic effects of plant-derived compounds are linked to acetylcholinesterase (AChE) inhibition, the modulation of gamma-aminobutyric acid (GABA), and octopamine (OA) receptors (GABAr and OAr, respectively). Essential oils are a complex mixture of green chemicals, mainly terpenoids, that act through these mechanisms, resulting in the overstimulation or paralysis of the insect’s nervous system [139]. Studies using Ag NPs evaluate the expression of glutathione S-transferase (GST) in midges, pointing to the importance of GST genes protecting against oxidative stress induced after exposure to Ag NPs [140].

Exposure to Ag NPs synthesized with *Alchornea cordifolia* leaf extract triggered behavioral and morphological changes in larvae, with reduced swimming [109]. *L. siceraria* ZnO NPs caused the loss of body hair, the decomposition of the outer cuticle’s epithelial layer, and the disintegration of abdominal structures and midgut on *A. stephensi* third instar larvae [99]. In addition, the abnormal restless movement of adult *A. stephensi* upon exposure to high concentrations of nanoformulated *Heliotropium indicum* extract was observed [126]. The histopathological analysis of fourth instar larvae of *A. culicifacies* exposed to *Cedrus deodara* essential oil pectin nanocapsules revealed severe morphological deformities and damage to muscle, adipose tissue, and epidermal cells. *C. deodara* essential oil has been reported to possess insecticidal activities attributed to the presence of compounds like terpenes, terpenoids, and sesquiterpenes, such as β-himachalene and α-himachalene [111]. *H. sprengelianum* essential oil components had different efficacy levels, lavandulyl acetate (LC50: 4.17 μg/mL) being more lethal to *A. subpictus* larvae than bicyclogermacrene (LC50: 10.3 μg/mL) [49]. Also, α-humulene and β-elemene isolated from *Syzygium zeylanicum* essential oil were highly toxic to *A. subpictus* larvae with LC50 6.19 and 10.26 μg/mL, respectively, which is even lower than the complete essential oil, which had LC50 83 μg/mL [141]. Carvacrol and terpinen-4-ol from *Origanum vulgare* essential oil were more effective than whole oil against *A. subpictus*, with an LC50 of 24.06 and 47.73 μg/mL, and *A. stephensi*, with an LC50 of 21.15 and 43.27 μg/mL, respectively [142]. *Plectranthus barbatus* essential oil’s major components, eugenol, α-pinene, and β-caryophyllene, were more effective against *A. subpictus* than whole extract, presenting an LC50 of 25.45, 32.09, and 41.66 μg/mL, respectively [143]. The main components from *Hedychium larsenii* essential oil, ar-curcumene and epi-β-bisabolol, were toxic against *A. stephensi* larvae (LC50: 10.45 and 14.68 µg/mL) and had oviposition-deterrent activity [144].

In field or semi-field experiments, it is hypothesized that the lethal effect of plant extracts may be due to the formation of an oil layer in the aqueous medium that blocks proper water oxygenation and allows for the active penetration of phytochemicals in larvae [110]. The water quality and compounds also implicate mosquito habits, such as egg-laying, because female gravid mosquitoes are attracted to breeding sites according to olfactory cues, such as pheromones and environmental odors, with species-dependent preferences [145]. *A. coluzzii* were more attracted to sites with early stages larvae [146] and *A. gambiae* had a preference to water sites treated with nonane, a pheromone produced by larvae [147]. Plant extracts may prevent oviposition due to their larvicidal effects, thus eliminating a mosquito attraction factor. Conversely, volatile plant compounds from natural habitats can attract mosquitoes and favor egg-laying, such as limonene, β-pinene, β-elemene, and β-caryophyllene detected in preferred *A. gambiae* breeding sites [148]. In this case, we hypothesize that the oviposition-deterrent activity by nanoformulations is due to the stability of these compounds in the water and the higher capacity for penetrating the insect’s body, causing toxicity.

The disadvantage of inorganic nanoparticles is the metal ion’s toxicity to the environment, highlighting the demand for an eco-friendly and modern intervention. For that reason, NPs synthesized from biomolecules is a relevant tool in fighting malaria whilst causing minimal impact on the environment. To summarize, the effects and possible action mechanisms are depicted in Figure 2, and we hypothesize that the toxicity of green NPs may be caused by the known effects of phytochemicals, which play a part in the functionality of NPs (as seen in Figure 1).

## 6. Antimalarial Effects of Green NPs

Multiple studies have evaluated the plant extracts potential to fight malaria at its many fronts, parasite elimination, or vector control. Regarding the vector population control, the main parameters assessed were the induction of larvae, pupae, and adult mortality, the impediment of oviposition, and egg hatchability. The main effects of green-synthesized NPs against mosquitoes and parasites are illustrated in Figure 3.

Among the studied vector species are *A. stephensi* and *A. subpictus*. *A. stephensi* is an essential vector of both *P. falciparum* and *P. vivax*, and has been documented to spread to other geographic areas, especially urban environments, where it can thrive, posing a challenge to vector control [1]. Thus, the use of natural products in these settings would be beneficial. Using plant extracts to synthesize NPs increases the mortality rate of mosquitoes in all developmental stages, and this effect is obtained using lower concentrations of the extract. For example, Ag NPs synthesized with *Euphorbia hirta* [124], *Nerium oleander* [125], *U. lactuca* [97], *Codium tomentosum* [114], *M. sylvestris* [108], *Holostemma adakodien* [81], *H. mystax* [104], *Quisqualis indica* [130], *Nicandra physalode* [131], *Gmelina asiatica* [132], *Musa paradisiaca* [149], *Drypetes roxburghii* [150], *Vinca rosea* [151], *Cocos nucifera* [112], *Pergularia daemia* [152], *L. siceraria* ZnO NPs [99], and *C. citratus* Au NPs [100] had the highest larvicidal and pupicidal effect against *A. stephensi*, in comparison to the crude plant extract. *S. xanthocarpum*-synthesized Ag NPs showed significant larvicidal effects up to 72 h after *A. stephensi* larvae exposure [105]. Also, against larvae of *A. subpictus*, Ag NPs synthesized with extracts of diverse plant species, such as *Zornia diphylla* [127], *Mussaenda glabra* [128], *Anisomeles indica* [129], *Barleria cristata* [119], *Ammannia baccifera* [120], *Nelumbo nucifera* [153], *Tinospora cordifolia* [154], and *Mimosa pudica* [155], and *Vitex negundo* ZnO NPs [117] were more toxic than the crude leaf extract, presenting lower lethal concentrations. The LC50 of green NPs and the affected mosquito species are summarized in Table 2.

A eucalyptus nanoemulsion had larvicidal activity against *A. stephensi* in laboratory and semi-field conditions. The larvae population had low density for up to 6 days in breeding sites sprayed with the nanoemulsion. In contrast to the breeding sites treated with regular eucalyptus oil, which had increased larvae after only two days [118]. In laboratory conditions, *Aristolochia indica* leaf extract-synthesized Ag NPs were toxic against *A. stephensi* larvae and pupae, with an LC50 ranging from 3.94 to 15.65 ppm, respectively. In field experiments, the crude extract and NPs had similar larvicidal activity, reaching about 50% larvae mortality in 24 and 100% in 72 h [110]. *Artemisia dracunculus* essential oil nanoemulsion and nano gel were toxic to *A. stephensi* larvae. The nanogel preparation was more lethal than the nanoemulsion, with an LC50 of 6.6 and 13.5 μg/mL, respectively [82]. Nanoliposomes containing *A. dracunculus* essential oil showed more significant larvicidal effect against *A. stephensi* when compared to other *Artemisia* species, such as *A. annua* and *A. sieberi,* even though all three presented toxic effects to larvae. *A. dracunculus* essential oil nanoliposomes caused 100% mortality at 50, 100 and 200 μg/mL, while *A. annua* reached this rate only at 200 μg/mL, and *A. sieberi* caused 77% [156]. On the other hand, *Acroptilon repens* essential oil nanoemulsion showed low larvicidal activity against *A. stephensi* because LC50 and LC90 values were 7 and 35 ppm, respectively [157].

*C. limon* bimetallic AuPd NPs showed larvicidal activity against *A. stephensi*. In the same study, larval mortality of *A. aegypti* was also assessed and proven to be lower because a 100% mortality rate was seen for the first instar larvae of *A. stephensi* but not *A. aegypti* [98]. Likewise, *A. cordifolia* Ag NPs were more lethal to *A. stephensi* larvae than other mosquito species [109]. Conversely, *Annona squamosa* Ag NPs LC50 values were higher for *A. stephensi* than *A. aegypti* and *C. quinquefasciatus* [122].

*Mangifera indica*-synthesized TiO_2_ NPs effectively controlled *A. subpictus* larvae. Even at the lowest concentration of 5 mg/L, TiO_2_ NP-exposed larvae had a higher mortality rate compared to TiO(OH) solution (TiO_2_ NPs: 37%; TiO(OH): 11%). TiO_2_ NPs reached the maximum mortality rate at 25 mg/L, whereas TiO(OH) solution had 89% mortality at the same concentration [84]. *I. frutescens*-synthesized Ag NPs showed a dose-dependent toxic effect on larvae of *A. subpictus*. The LC50 for Ag NPs was 14.22 μg/mL, and for the aqueous extract, it was 185.83 μg/mL. The highest mortality rate, 100%, was only obtained by NPs at a concentration of 35 μg/mL, while the crude extract reached 99.2% mortality at 450 μg/mL [101]. *A. nilagirica* Ag NPs had significant toxic effects against *A. stephensi* first–fourth instar larvae and pupae, with LC50 ranging from 0.343 to 0.05% after 24 h of exposure at a 0.25% concentration level. The *A. nilagirica* Ag NPs reached 100% mortality of late instars and pupae at 0.20% concentration. The plant aqueous extract did not reach these mortality rates even at the highest concentration of 1% [121], in addition to *M. charantia* leaf extract-synthesized TiO_2_ NPs [106] and ZnO NPs [123]. TiO_2_ NPs were toxic to first–fourth instar larvae and pupae of *A. stephensi*, with an LC50 ranging from 2.5 to 5.04 mg/L. The ZnO NPs had a larvicidal effect against *A. subpictus*, with an LC50 of 5.42 mg/L. Both *M. charantia* NPs had higher toxicity than the plant aqueous extract. *P. aquilinum* Ag NPs showed high toxicity to larvae, pupae, and adult *A. stephensi* in laboratory and field conditions. The NP treatment reduced adult mosquitoes’ longevity and fecundity. Applying plant-based treatments in water reservoirs reduced the larval population to zero within 72 h of exposure [96]. *R. ellipticus*-synthesized Ag NPs also displayed ovicidal (LC50: 60 μg/mL) and adulticidal (LD50: 21.10 μg/mL) effects and prevented 89% oviposition at 60 ppm concentration [102].

Also, researchers synthesized essential oil from *C. deodara* into pectin nanocapsules and applied the solution to cotton-bag fibers in water containing *A. culicifacies* third instar larvae. Within five days, the larval population was reduced by 90%, reaching a 98% reduction after 28 days [111]. *Mimusops elengi* Ag NP has also displayed toxicity against mosquito larvae, pupae, and adult mosquitoes at ultra-low doses. Intriguingly, the ultra-low dosages of green-synthesized Ag NP reduced the motility of mosquito larvae, consequently boosting the efficiency of natural mosquito predators, like mosquitofish [115]. *Datura metell* leaf extract that contains alkaloids and atropine, used for medicinal purposes [158], and *Aloe vera* extract [159] Ag NPs showed acute toxicity against larvae and pupae of *A. stephensi*, in addition to enhancing predation to larval populations. In field conditions, *A. vera* Ag NP led to a larval reduction of 97.7% after 72 h [159]. *Annona muricata* extract possesses several properties, such as anti-viral, anti-cancer, anti-fungal, and especially larvicidal activity. Ag NPs from *A. muricata* showed toxicity against *A. stephensi* third instar larvae with LC50 and LC90 values of 15.28 and 31.91 μg mL (−1) [160].

In addition, *Chrysanthemum indicum*, a flower rich in pyrethrin, has gained widespread recognition for its insecticidal properties attributed to its impact on membrane integrity, causing damage to both lipid and aqueous components of the gill membrane. Notably, *C. indicum*-derived Ag NPs exhibited remarkable mortality rates when tested against larvae and pupae of *A. stephensi* [161]. The peanut (*Arachis hypogaea*) peel extracts were utilized to synthesize Ag NP and evaluated against fourth instar larvae of *A. stephensi*. When exposed to a concentration of 15 mg/L, a 100% mortality rate was observed among the larvae within 24 h, accompanied by significant morphological changes in the cuticular membrane [116]. A similar outcome was observed with the Ag NPs derived from the aqueous leaf extracts of *Leucas aspera* and *Hyptis suaveolens*, where a mortality rate of 100% was achieved at a concentration of 10 mg/L [162]. Biosynthesized Au NPs of *Couroupita guianensis* flower extract were used in water storage reservoirs on larval populations of *A. stephensi*. The Au NP flower extract had larvicidal and pupicidal effects, and the adulticidal activity showed 95% mortality. The predation efficiency of *Aplocheilus lineatus* after treatment with ultra-low dosages of *C. guianensis* Au NP was enhanced (96.04%) [113].

A few reports indicated that green-synthesized NPs may also have toxic effects against malaria parasites in cellular and animal models. The most studied *Plasmodium* strains are the *P. falciparum* chloroquine-sensitive (CQs) strain 3D7, and chloroquine-resistant (CQr) strains RKL9 and INDO. An early in vivo report assessed the potential of orally administered palladium (Pd) NPs synthesized from *Eclipta prostrata* in mice infected with the *P. berghei* CQs strain NK 65. Pd NPs were more effective in reducing parasitemia than the aqueous extract [84]. *A. cordifolia* Ag NPs were more toxic to 3D7 and RKL9 strains than crude extract, with IC50 values 8.05 and 20.27 μg/mL for 3D7, 10.31 and 32.55 μg/mL for RKL9, respectively [109]. Ag NPs synthesized with *Azadirachta indica* inhibited parasite growth. The IC50 (µg/mL) values for *A. indica* bark Ag NPs against 3D7 and RKL9 strains were 8 and 7.8, respectively. *A. indica* leaf Ag NPs had an IC50 of 9.2 for 3D7 and 11.1 for RKL9. Both NPs were effective at lower concentrations than the aqueous extract [41]. Antiplasmodial effects were also seen for *L. siceraria* ZnO NPs, with parasitemia suppression at IC50 4.32 µg/mL, comparable to chloroquine with IC50 2.5 µg/mL [99]. Compared to the crude leaf extract, *M. charantia*-synthesized TiO_2_ NPs were more effective in the antiplasmodial assay against 3D7 and INDO strains [106]. *C. tomentosum* Ag NPs inhibited the parasitemia of 3D7 and INDO strains, with an IC50 of 72.45 and 76.08 μg/mL, respectively. The treatment was similar to chloroquine, with an IC50 of 80 (CQs strain) and 85 μg/mL (CQr strain) [114]. Interestingly, Ag NPs and crude extract from *U. lactuca* [97] and *P. aquilinum* [96] showed higher toxicity to 3D7 and INDO parasite strains than chloroquine. The C. guianensis extract Au NPs showed a higher growth inhibition than chloroquine against 3D7 and INDO strains [113]. Also, the *L. siceraria* ZnO NPs exhibited the potent inhibition of hematin formation [99]. *E. eugeniae* Ag NPs showed higher toxic activity in antiplasmodial assays than chloroquine against *P. falciparum* [66]. After the infection of red blood cells, *P. falciparum* digests hemoglobin, generating a free heme group, which is toxic to the parasite. Within its digestive vacuole, *Plasmodium* converts the heme group into hemozoin, also called β-hematin, a non-toxic crystallized molecule [163]. So, the inhibition of hemozoin formation is crucial in the research for novel antimalarials. Thus, pilling evidence shows the potential use of green NPs to fight malaria.

## 7. Conclusions

While it is well-documented in the literature, there needs to be more research discussions regarding the relevance of green-synthesized NPs in malaria prevention and its impact on the vector. Many studies have indeed explored its potential in treatment, but there needs to be more exploration of its role in vector prevention. In this paper, our primary objective was to conduct a comprehensive review of previously synthesized nanoparticles that are potentially used in controlling malaria transmission. By addressing this gap in research, we aim to contribute valuable insights into malaria prevention. Nanotechnology is leading with the most recent research for better drug delivery and the development of novel tools. The nanoformulation of compounds enables low concentrations compared to the bulk chemicals, thereby reducing toxicity. Indeed, nanotechnology characteristics and materials are an essential topic in discussion since they can influence the biological impact and behavior of the NPs. There are disadvantages to inorganic NPs, so the biogenic synthesis of NPs has been extensively reported in the literature, especially regarding the use of plant extracts in the synthesis of inorganic NPs with potent activity against the malaria vector and parasite. This green synthesis of NPs allows the bio compounds plant polyphenols to enhance the effects of metallic NPs, causing an overall toxic impact on all developmental stages of the mosquito vector, reflected in larval mortality, reduced oviposition, impaired feeding habit, and morphological and behavioral changes. In addition, NPs synthesized with plant extracts show antiplasmodial effects by inhibiting parasite growth. Furthermore, choosing interventions against malaria should take into account equal access, sustainability, and human health impact.

## 8. Future Perspectives

The lack of evidence of the plant extracts’ mechanism of action becomes a significant issue in this review. The complex chemical composition of plants can hinder the emergence of resistance because the toxic compounds can vary between plant species. Although plant-derived nanoformulations are not currently used, we advocate for robust research regarding the long-term bioavailability of green nanoparticles, potential toxicity, production scaling, and administration feasibility.

## Figures and Tables

**Figure 1 pharmaceutics-16-00699-f001:**
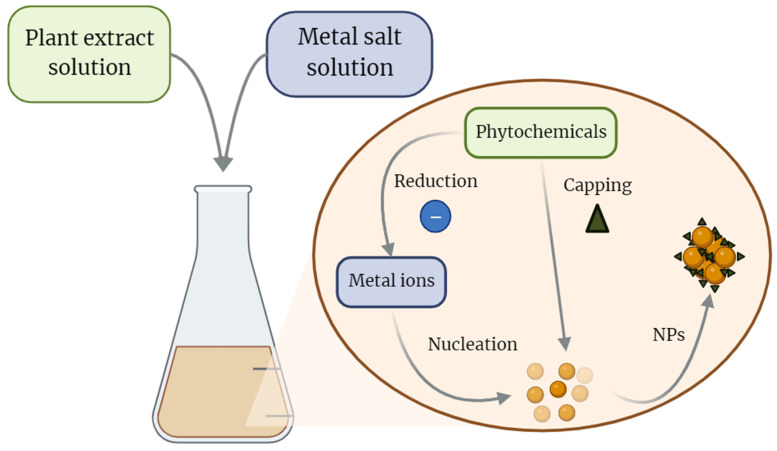
Schematic diagram for the biosynthesis of nanoparticles (NPs) via a green route using plant extract. A plant extract is a rich mixture of various metabolites and reductive biomolecules, each playing a pivotal role in the intricate process of reducing metal ions. Biomolecules stand out for their ability to transfer electrons to metallic ions, acting as potent reducing agents that instigate the ions’ reduction. The consequence of this reduction process is the development of structured patterns in the form of a crystalline formation known as a nucleus. The initiation of nucleation involves the aggregation of monomers in supersaturated systems. This crucial step establishes the foundation for the subsequent growth of these nuclei, leading to the formation of metal nanoparticles. As these atomic nuclei form, they provide a foundational platform for the reduced ions, facilitating their gradual deposition onto the surface. This deposition process significantly contributes to the enlargement of the particles. Biomolecules, in addition to their role as reducing agents, also serve as capping agents. They bind to the particle surface, acting as molecular guardians that effectively stabilize the size of the particles.

**Figure 2 pharmaceutics-16-00699-f002:**
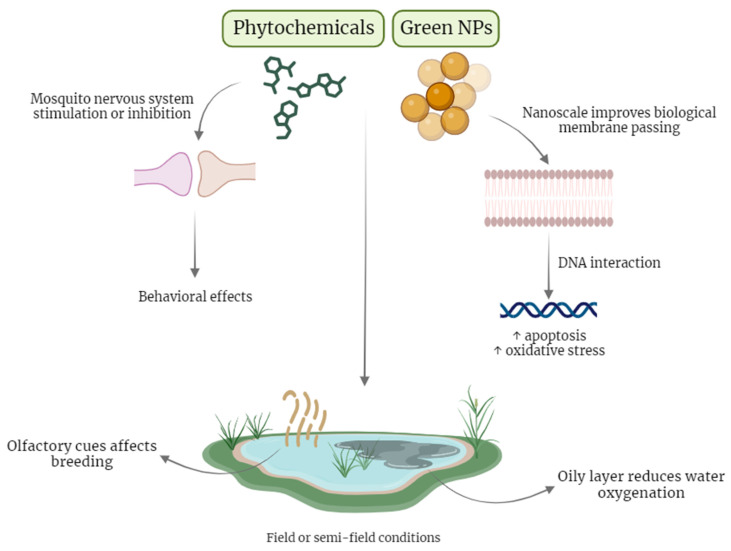
Action mechanisms of plant extracts and plant-derived nanoparticles. In the left panel, we summarize the phytochemicals’ effects. These molecules are recognized for their larvicidal and mosquitocidal properties, modulating the mosquito nervous system, by inducing either overstimulation or inhibition. This neuro-modulation results in observable behavioral changes, such as the reduced swimming and restless movement of larvae. In the right panel, effects of green nanoparticles (NPs) are illustrated. NPs with their optimal size effortlessly penetrate the membranes of mosquitos. Their mechanism of action involves interactions with DNA, contributing to enhanced apoptosis and oxidative stress within the target organisms. Notably, when deployed in field or semi-field conditions, these nanoparticles form an oily layer, a phenomenon correlated with a reduction in water oxygenation. Furthermore, both phytochemicals and green NPs influence factors such as the attraction of female mosquitoes to breeding sites, which are guided by olfactory cues susceptible to water quality. The effects of green-synthesized NPs may be due to the known effects of biomolecules present in plant extracts.

**Figure 3 pharmaceutics-16-00699-f003:**
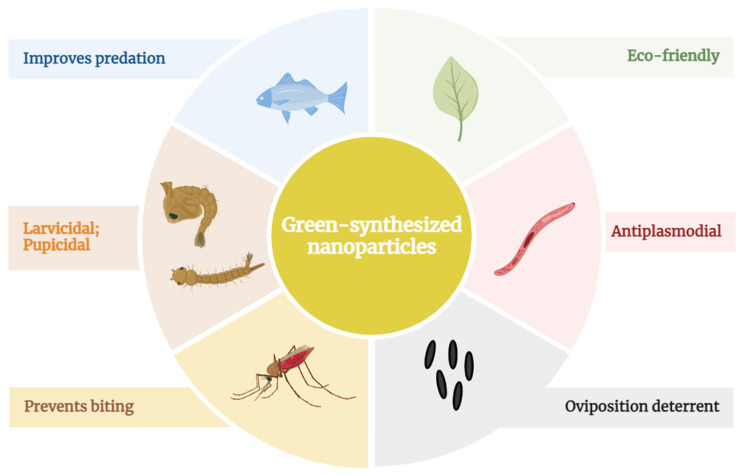
Effects of green-synthesized NPs on malaria prevention. NPs synthesized with plant extracts are toxic to early developmental stages of the malaria mosquito vector, *Anopheles*, acting against all four larvae stages (instar) and pupae, preventing the emergence of adult mosquitos capable of disease transmission. These nanoformulations also show adulticidal effects, impacting mosquito behavior and causing morphological damage. Furthermore, some studies showed that plant-extract NPs reduced the feeding of mosquitoes by preventing the biting habit. Oviposition is hindered in water treated with the nanoformulated plant extracts, thus reducing breeding site suitability. Green-synthesized NPs also improve the feeding activity of mosquito predators, lowering the survival chances of mosquito larvae and pupae. These formulations are eco-friendly because predators are not harmed. In addition, nanotechnology emphasizes the well-known antiparasitic activity of plant extracts. The antiplasmodial effects of plant-derived nanoformulations are parasitemia suppression by increased parasite mortality and metabolic impairment. Therefore, plant-derived nanoformulations are a potential tool to fight malaria.

**Table 2 pharmaceutics-16-00699-t002:** Half-lethal concentration (LC50) of nanoformulated plant products and bulk formulations in late developmental stages (third–fourth instar and pupae) of *Anopheles* species after 24 h of exposure.

Plant Species	Type of Nanoparticle	Mosquito Species	Mosquito Stage	LC50 of Nanoformulation	LC50 of Bulk Extract	Reference
*Eucalyptus globulus*	Nanoemulsion	*A. stephensi*	Larvae	80.8730 ppm	122.8343 ppm	[118]
*Mangifera indica*	TiO_2_	*A. subpictus*	Larvae	7.72 mg/L	49.45 mg/L	[84]
*Ichnocarpus frutescens*	Ag	*A. subpictus*	Larvae	14.22 μg/mL	185.83 μg/mL	[101]
*Citrus limon*	AuPd	*A. stephensi*	Larvae	5.12 mL/L	-	[98]
*Rubus ellipticus*	Ag	*A. stephensi*	Larvae	12.50 μg/mL	167.19 μg/mL	[102]
*Naregamia alata*	Ag	*A. stephensi*	Larvae	12.40 μg/mL	165.15 μg/mL	[103]
*Alchornea cordifolia*	Ag	*A. stephensi*	Larvae	10.67 μg/mL	53.15 μg/mL	[109]
*Solanum xanthocarpum*	Ag	*A. stephensi*	Larvae	9.927 ppm	1764.528 ppm	[105]
*Barleria cristata*	Ag	*A. subpictus*	Larvae	12.46 μg/mL	124.27 μg/mL	[119]
*Malva sylvestris*	Ag	*A. stephensi*	Larvae	10.33 μg/mL	143.61 μg/mL	[108]
*Ammania baccifera*	Ag	*A. subpictus*	Larvae	29.54 mg/L	257.61 mg/L	[120]
*Artemisia annua*	Nanoliposomes	*A. stephensi*	Larvae	90 μg/mL	-	[156]
*Artemisia sieberi*				140 μg/mL		
*Artemisia dracunculus*				23 μg/mL		
*Artemisia nilagirica*	Ag	*A. stephensi*	Larvae	0.141%	0.224%	[121]
			Pupae	0.050%	0.066%	
*Valoniopsis pachynema (algae)*	CdS	*A. stephensi*	Larvae	29.429 μg/mL	334.084 μg/mL	[107]
			Pupae	31.905 μg/mL	396.868 μg/mL	
		*A. sundaicus*	Larvae	22.496 μg/mL	296.922 μg/mL	
			Pupae	25.009 μg/mL	311.860 μg/mL	
*Vitex negundo*	ZnO	*A. subpictus*	Larvae	2.48 mg/L	36.89 mg/L	[117]
			Pupae	3.63 mg/L	45.76 mg/L	
*Aristolochia indica*	Ag	*A. stephensi*	Larvae	10.48 ppm	490.31 ppm	[110]
			Pupae	15.65 ppm	565.02 ppm	
*Annona squamosa*	Ag	*A. stephensi*	Larvae	2.12 ppm	-	[122]
			Pupae	3.74 ppm		
*Ulva lactuca*	Ag	*A. stephensi*	Larvae	5.261 ppm	32.692 ppm	[97]
			Pupae	6.860 ppm	37.603 ppm	
*Momordica charantia*	ZnO	*A. stephensi*	Larvae	5.42 mg/L	52.86 mg/L	[123]
	TiO_2_		Larvae	3.43 mg/L	85.35 mg/L	[106]
			Pupae	5.04 mg/L	96.09 mg/L	
*Pteridium aquilinum*	Ag	*A. stephensi*	Larvae	18.45 ppm	395.12 ppm	[96]
			Pupae	31.51 ppm	502.20 ppm	
*Euphorbia hirta*	Ag	*A. stephensi*	Larvae	27.89 ppm	197.40 ppm	[124]
			Pupae	34.52 ppm	219.15 ppm	
*Nerium oleander*	Ag	*A. stephensi*	Larvae	33.99 ppm	369.96 ppm	[125]
			Pupae	39.55 ppm	426.01 ppm	
*Codium tomentosum (algae)*	Ag	*A. stephensi*	Larvae	29.6 ppm	410.7 ppm	[114]
			Pupae	40.7 ppm	487.1 ppm	
*Cymbopogon citratus*	Au	*A. stephensi*	Larvae	31.466 ppm	362.292 ppm	[100]
			Pupae	38.327 ppm	434.649 ppm	
*Hugonia mystax*	Ag	*A. stephensi*	Larvae	14.45 μg/mL	162.66 μg/mL	[104]
*Lagenaria siceraria*	ZnO	*A. stephensi*	Larvae	56.46 ppm	261.67 ppm	[99]
*Heliotropium indicum*	Ag	*A. stephensi*	Adult	26.712 μg/mL	111.680 μg/mL	[126]
*Zornia diphylla*	Ag	*A. subpictus*	Larvae	12.53 μg/mL	61.23 μg/mL	[127]
*Mussaenda glabra*	Ag	*A. subpictus*	Larvae	17 μg/mL	81 μg/mL	[128]
*Anisomeles indica*	Ag	*A. subpictus*	Larvae	31.56 μg/mL	108.98 μg/mL	[129]
*Holostemma adakodien*	Ag	*A. stephensi*	Larvae	12.18 μg/mL	185.79 μg/mL	[81]
*Quisqualis indica*	Ag	*A. stephensi*	Larvae	12.52 μg/mL	185.98 μg/mL	[130]
*Nicandra physalodes*	Ag	*A. stephensi*	Larvae	12.39 μg/mL	202.82 μg/mL	[131]
*Gmelina asiatica*	Ag	*A. stephensi*	Larvae	22.44 μg/mL	113.53 μg/mL	[132]
*Couroupita guianensis*	Au	*A. stephensi*	Larvae	24.57 ppm	307.72 ppm	[113]
			Pupae	28.78 ppm	363.25 ppm	

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
