# Peer review of "Tiny Green Army: Fighting Malaria with Plants and Nanotechnology"

_pharmaceutics, 2024, doi:10.3390/pharmaceutics16060699_

Round 1

Reviewer 1 Report

Comments and Suggestions for Authors

1. Based on the introduction to the article, several areas can be noticed that could benefit from further elaboration or clarification:

·         Significance of the COVID-19 Pandemic: It would be worthwhile to provide a more detailed discussion of how the COVID-19 pandemic has impacted the fight against malaria. The authors mention limitations in the supply of Insecticide-Treated Nets (ITNs) and disruptions in malaria diagnosis and treatment, but they could provide more data and evidence to support these claims.

·         Role of Remedial Measures: In the context of malaria management, a more detailed discussion of the role of remedial measures, such as preventive measures and drugs, would be beneficial. The authors can provide more information on the availability of these measures in different regions of the world and the challenges associated with their distribution and accessibility.

·         Mechanisms of Parasite and Mosquito Resistance: Describing the resistance mechanisms of Plasmodium parasites to drugs and Anopheles mosquitoes to insecticides is crucial. The authors can expand on this topic, discussing specific resistance mechanisms and methods for monitoring and managing them.

·         New Technologies and Research: The introduction suggests the need for further research and innovation in the field of malaria control. The authors can elaborate on this point by providing examples of new technologies or research approaches that are currently being developed or studied in the context of malaria control. 

2. The chapter "Green-nanotechnology" provides extensive technical information about nanotechnology and discusses the applications of nanosystems such as nanoparticles (NPs), nanoemulsions, and nanogels in the context of medicine and malaria control. Overall, the text is well-written and conveys important information. However, there are areas that could benefit from improvement or expansion:

·         Precise Sources and Examples: In some places in the article, it might be useful to provide precise sources or examples to emphasize and document the information. For example, when mentioning the "biogenic synthesis approach" and the benefits of using plant extracts for synthesizing Ag NPs, the authors could provide more specific examples of such research or products.

·         Clarity in Process Descriptions: Some sentences are technical and may be difficult for readers without an advanced background in nanotechnology to understand. The authors can attempt to explain processes more clearly, such as "nucleus" and "capping agents," to facilitate understanding for readers outside the field.

·         Elaboration of Concepts: Several important concepts are introduced, such as the role of biomolecules in the reduction and encapsulation of NPs, but these concepts could be more elaborated to explain their significance and relevance to the article's topic.

3. The chapter "Plants and Nanoformulations" provides extensive information about the use of plants and nanoformulations in the context of malaria control. Overall, the text is well-written and conveys important information, but some revisions and additions could be considered:

·         Clarity in Descriptions: In some places, especially in descriptions of research results regarding various plants and substances, it might be more helpful to present data such as LC50 (half-lethal concentration) values and other parameters more clearly. Clarity in descriptions will help readers better understand the research results.

·         Collaboration with Other Organisms: The text mentions the harmlessness of certain non-target organisms, such as fish and earthworms. However, more information could be provided about why these organisms are not at risk or the benefits of their presence in the ecosystem.

·         Relevance to Malaria: In some places, particularly in the context of nanoformulations, it may be helpful to explain how these techniques or substances are related to the fight against malaria. Explaining specific mechanisms of action and benefits in the context of malaria can be helpful for readers.

4. The chapter "Malaria prevention strategies" discusses various strategies for preventing malaria, especially those related to vector control and chemotherapy. Here are some comments regarding this chapter:

·         Precise Sources and Numbers: In some places, especially in descriptions of research results regarding the toxicity of plants and nanomaterials, it might be useful to provide more specific numbers and sources to support the presented information. This will help readers better understand and evaluate the results.

·         Concrete Examples: In describing various alternative malaria prevention strategies, more concrete examples of plants or nanomaterials that have been studied and have shown promising results in malaria control could be provided.

·         Emphasize Ecological Significance: While the text mentions eco-friendly alternatives, it would be beneficial to emphasize the significance of these strategies for environmental protection and maintaining ecological balance. More information could also be provided about potential negative impacts on ecosystems.

5. The chapter "Plant extracts and green NPs action mechanisms" focuses on the mechanisms of action of nanoparticles and plant extracts in combating malaria-transmitting mosquitoes. Here are some comments regarding this chapter:

·         Clarity on Toxicity: In some places, especially in the description of the toxicity of nanoparticles and plant extracts, more information about potential negative effects on the environment and human health could be provided to explain why these mechanisms are important.

·         Understandable Examples: In describing various mechanisms of action, using more understandable examples or analogies that help readers better understand biological processes, such as interactions with proteins or DNA, could be beneficial.

6. The conclusion section is well-written and provides important information about the role of nanotechnology and green nanosystems in the fight against malaria. Here are some comments on these conclusions:

·         Conciseness: The text is relatively long. It's important to ensure that the conclusions are as concise as possible and focused on the key points. This helps readers better understand the main message.

·         Summarize Key Points Earlier: Consider summarizing the main points earlier in the text rather than in the final sentences. Providing a clear overview of what is discussed can help readers grasp the main points.

·         Context: Consider adding brief context to explain why these conclusions are important and what significance they hold in the context of malaria control.

Encouragement for Further Research: You may consider adding a brief section that encourages further research or continued work in this field.

Overall, the conclusions are generally good, but they could benefit from a more concise summary and possibly additional context to help readers better understand the significance of these conclusions.

Author Response

Please find the answer in the attached document. 

Reviewer 2 Report

Comments and Suggestions for Authors

1.       The manuscript should be carefully proofread in terms of typos and language.

2.       The attractive tittle of the manuscript were missing.

3.       If so, the authors advised to include the recent epidemiological data of malaria.

4.       Line 134: Check the statement, "Biomolecules exhibit the capacity to transfer electrons to metallic ions, functioning as potent reducing agents that initiate the ions' reduction." If so, what about  caping

5.       The authors mentioned hydrogel and nanoemulsion; unfortunately, I didn't see any literature about its anti-malarial activity.

6.       Some references, like ref No: 1, 2 8, are repeatedly cited. Please check it,

7.       Green synthesis of silver nanoparticles using plant sources is well documented; why is this paper so crucial for the scientific community? Please justify, and the authors need to articulate this in the paper too. Check this paper: https://www.sciencedirect.com/science/article/pii/S0882401018318175; https://doi.org/10.1016/j.mtcomm.2023.105652

Author Response

Please find the answers in the attached document.

Reviewer 3 Report

Comments and Suggestions for Authors

The manuscript entitled "Green Tiny Army: Nanoformulated plant extracts in the fight against malaria" Title, abstract and overall rationale of work is written satisfactory. Still, there are major concerns, which needs to be addressed before publication.

1) Abstract is written well and concise way. Only one suggestion to incorporate here sp. Should be write complete like species.

2) Keywords: Some important keywords are missing author must be add like Plasmodium, ACT and others

3) The introduction is written too much details and I recommend to reduce this section and more focus in your green nanoformualtion plant extract to fight against malaria parasite.

4) In this section Green-nanotechnology, author explained deep about the NP and I suggest author to give one pictorial (figure) view here to show clear picture.

5) Line no.182-185 author stated about the condition of the drug during field condition and I am not completely agree with these sentences and author need to revise.

6) Section 4: author repeated the information again specially first paragraph and I recommend to remove this repetition.

7) Section 5 author must be provide one figure to show clear mechanism of the action of these NP against malaria.

8) Section 6 author need to short this section because they already shown all these explanation in the table 2. Kindly reduce the written part and make more attractive.

9) Author need to incorporate important/significance and future prospective of this review in the conclusion section.

10) All references is satisfactory and cite properly.

Comments on the Quality of English Language

English is written very good.

Author Response

(The authors gave the same response as above.)

Round 2

Reviewer 1 Report

Comments and Suggestions for Authors

I accept this article in its present form. Thank you for your changes. 

Author Response

Thank you. Your comments help us to improve the overall quality of the manuscript. 

Reviewer 3 Report

Comments and Suggestions for Authors

The authors have addressed all the concerns raised in the previous version of the manuscript and the quality has much improved after incorporating required modifications. Therefore, the manuscript may be considered for publication in this Journal.

Author Response

Dear Reviewer,

Thank you. Your comments help us to improve the overall quality of the manuscript.